# Public Understanding of Ignorance as Critical Science Literacy

## Fabien Medvecky

Centre for Science Communication, University of Otago, Dunedin 9016, New Zealand; fabien.medvecky@otago.ac.nz

**Abstract:** We are largely ignorant. At least, there are many more things we are ignorant of than knowledgeable of. Yet, the common perception of ignorance as a negative trait has left it rather unloved in debates around making knowledge public, including science communication in its various guises. However, ignorance is a complex and essential part of science; it performs a number of legitimate roles, and is performed in a range of legitimate ways within science. In this paper, I argue that it is vital to understand when ignorance is an appropriate, legitimate part of the scientific process, and when ignorance is misused or abused in science. I argue that understanding ignorance is a central aspect of public understanding of science, especially in terms critical science literacy. Critical science literacy argues that more than simply an understanding of scientific facts and processes, a key component of what scientific literacy should aim for is an understanding of the tacit knowledge of science. I present a typology of ignorance and argue that fostering a greater public understanding of ignorance is a rarely acknowledged, yet essential, aspect of making science public, and that it is a challenge that those engaged in and committed to better public understanding of science should take very seriously.

**Keywords:** public understanding of science; critical science literacy; agnotology; ignorance; science communication

## 1. Introduction

In a strange paradox, we claim to live in a knowledge society [1,2], yet we are largely ignorant. At least, there are many more things we are ignorant of than knowledgeable of. Despite ignorance being ever so prominent in our relationship to knowledge, it is viewed, more often than not, as a bane. While there has been some limited interest in ignorance in various branches of scholarship [3,4], the common perception of ignorance as a negative trait has left it rather unloved in debates around making knowledge public, including science communication and its various cousins. We suffer an "ignorance of ignorance", or, as Ravetz eloquently put it, "ignorance-squared" [5]. Yet ignorance is a complex and essential part of science, from being a driver of curiosity to setting limits on expertise [6]. Ignorance performs a number of legitimate roles, and is performed in a range of legitimate ways within science. On the other hand, ignorance can also be misused and abused, intentionally or naively.

In this paper, I will make a case for the importance of the public understanding of ignorance to the public understanding of science. I will argue that understanding when ignorance is a legitimate part of the scientific process—and when ignorance is misused or abused in science—is central to understanding science in terms of traditional public understanding of science, but especially if we think in terms of critical science literacy.

Critical science literacy argues for more than simply an understanding of scientific facts and processes; it argues that an understanding of the tacit knowledge of science is a key component of what scientific understanding and literacy should aim for [7].

To make my case, I begin by defining the limits of ignorance, making the case that ignorance, far from being the curse we ought to break-free from in our lives, is our stock-standard underlying condition (so better we learn to live happily with it!). Drawing on the

existing scholarship, I then present a typology of ignorance, and pay special attention to its uses (and misuses) in science. In the third part of this paper, I make explicit what I take to be critical science literacy, locating it within the greater context of science communication and the public understanding of science. Lastly, I bring the former parts together to argue that fostering a greater public understanding of ignorance is a rarely acknowledged, yet essential, aspect of making science public, and is a challenge that those engaged in and committed to a better public understanding of science should take very seriously.

## 2. What Is Ignorance

If I were to call someone ignorant, they would likely be offended. Ignorance is usually considered a flaw in an individual. It is presented as the antithesis of knowledge, and knowledge is good. Being knowledgeable is a compliment. Knowledge is the light (as the World Bank put it [8]) that will dispel the darkness of ignorance [9]. Indeed, there seems to be little love for (and little research on) ignorance. As Proctor and Schiebinger, who coined the term "agnotology", claim: "Ignorance hides in the shadows of philosophy and is frowned upon in sociology" (2008). Yet even they mostly present ignorance in a bad light, opening their volume about the topic with a goal: "to explore how ignorance is produced or maintained in diverse settings, through mechanisms such as deliberate or inadvertent neglect, secrecy and suppression, document destruction, unquestioned tradition, and myriad forms of inherent (or avoidable) culturopolitical selectivity". Proctor and Schiebinger do not use the term agnotology as the study of ignorance in all its (gore and) glory; it is reserved for the study of culturally induced ignorance. When we study ignorance, it is usually the bad aspects we focus on.

Sure, not all ignorance is bad. Scholars have noted that ignorance can be strategic and helpful [10]. More relevant for this paper, it has also been noted that ignorance is an essential part of science [11]; it is what drives science and allows for discovery. There would be no search for an answer or solution to a scientific question if we were not initially lacking one—if we were not firstly ignorant.

Ignorance, at its most basic, is "not knowing". In particular, the focus in this paper is on the "not knowing" of facts, claims, information, and so forth in two ways: either lacking any knowledge or holding a false knowledge. This view assumes a broad definition of ignorance, which will be discussed as we proceed. However, there are some clarifications to get out of the way first. Someone might be referred to as being ignorant because they "ignore" social customs or trespass some moral or social code. This disregarding of norms is not (necessarily) a result of ignorance as "not knowing"—it sometimes is intentional non-compliance. Indeed, as Margaret Atwood noted: "Ignoring isn't the same as ignorance, you have to work at it" [12]. The distinction between ignorance and ignoring is one area where knowledge and ignorance are asymmetrical; you can be ignorant without actively ignoring, but "[f]or knowledge to be, there must have been, at some stage, an act of knowing" [13]. This paper will be on both ignorance (the state) and ignoring (the act), as the latter can also sometimes lead to ignorance, as will be discussed further below.

Thinking of ignorance in this way narrows the field, yet it takes nothing away from the enormity of our ignorance, both as individuals and collectively. Put simply, while each individual may have some limited, finite knowledge about a few topics, the volume of claims and propositions any individual is ignorant of is infinite. Put mathematically, at an individual level, our knowledge, relative to our ignorance, approaches zero. Collectively, we are not much better off. While there is much more collective knowledge (defined broadly in terms of complementary knowledge and embedded knowledge as the sum of the knowledge held, not just in the individuals, but in the collective artefacts, books, etc. [14]), there is still significantly more ignorance than knowledge. Worse still, knowledge (and science is an exemplar case here) does not so much dispel ignorance as open new spaces of ignorance; ignorance "increases with every state of new knowledge" [4]. The more we know, the more we know we do not know. Ignorance, it turns out, is our most

fundamental underlying condition. As humans, we are mostly ignorant, with few, rare glimpses of knowledge, and there are many ways we can carve up our ignorance

## 3. Typology of Ignorance

Unsurprisingly, it was the great sociologist of science, Merton, who first brought our attention to the functions of ignorance in our structure of knowledge, and specifically in science. He carved up ignorance into specified and unrecognized ignorance. Merton used *specified ignorance* to refer to "the express recognition of what is not yet known but needs to be known in order to lay the foundation for still more knowledge" [15] and noted that "*specified* ignorance is often a first step toward supplanting that ignorance with knowledge" [16]. *Unrecognized ignorance*, by contrast, is that which is left off our radars such that "as yesterday's uncommon knowledge becomes today's common knowledge, so yesterday's unrecognized ignorance becomes today's specified ignorance" [15]. For the sake of simplicity, I will refer to these as *recognized* and *unrecognized ignorance*, respectively.

We can further formalize the typology of ignorance by drawing our attention to what cannot be known. As Ungar explains, there is "ignorance that exists beyond the boundaries of knowledge, such as scientific ignorance about new or unknown phenomena, ranging from the aspects of the brain through to black holes" [17]. Wilholt distinguishes between what he terms conscious ignorance (Merton's *recognized ignorance*) and *deep ignorance*, where *deep ignorance* is the subclass of *recognized ignorance* that, while recognized, has no candidate answers [18]. An example might be knowing what it is like inside a black hole; we can ask the question and recognize our ignorance, but we also know we cannot answer it and are bound to, at least for now, remain ignorant. Wilholt further distinguishes between *unrecognized ignorance* (which he terms opaque ignorance) and *thoroughly opaque ignorance*, where *thoroughly opaque ignorance* is the subclass of *unrecognized ignorance* that is not only left off our radar, but could not even make it onto our radar, as we do not have the conceptual capacity to formulate the question [18]. An example here is Aristotle's ignorance of what it is like inside a black hole. Not only is there no answer to that question, but Aristotle also lacked the conceptual capacity to even make sense of the question, given the state of astronomy in classical Greece.

Ignorance has also received substantial treatment in formal systems that try to contain or manage our ignorance, especially in scientific and technical systems. In such settings, ignorance is treated along formal lines, in terms of probability assignments, risk-analysis, and formal decision-theoretic frameworks. Risk, uncertainty, and probabilities are well-respected and established family relations of ignorance [19]. These require some form of ignorance and can be seen as ways to manage our ignorance. We can think of risk and uncertainty as fundamentally about ignorance as to which state of affairs is true, and probabilities as a way of quantifying this ignorance. Note that while risks and uncertainty are related to ignorance, they are not equivalent to it. There is much to ignorance that lays beyond this. Indeed, the concept of ignorance is an integral part of decision theory and risk analysis, such as in "decisions under ignorance", where it is interpreted as a unique and "extreme state of uncertainty" defined as "a singular state of knowledge characterized by knowing nothing or having no reliable information about the phenomenon of interest" [20]. Though treating ignorance in this way continues the long-standing scientific narrative and ideal of managing or "replacing ignorance by knowledge, with little attention to the formation of a useful kind of ignorance" [15]. This paper, to some extent, hopes to redress this by giving attention to this "useful kind of ignorance".

Table 1 shows how the formal typology of the recognized/unrecognized typology of ignorance mentioned above relates to more recent social, agnotology typology, which is discussed below.

Since the mid-2000s, ignorance has taken a social turn, most notable since the publication of Proctor's and Schiebinger's *Agnotology* [21]. Rather than focusing on formal structures and divisions of ignorance, this recent turn focuses our attention on the socio-cultural aspects of ignorance. Much like the more formal work on ignorance can be thought

of as the ignorance counterpoint to epistemology, the work stemming from agnotology can be thought of the ignorance counterpoint to social epistemology [22]. Proctor and Schiebinger also offer a typology, but fitting the social move, one based on socio-cultural attributes. Agnotology distinguishes between the active construction of ignorance and passive construction of ignorance. Active constructions of ignorance are intentional forms of ignorance while passive constructions of ignorance are "the unintended by-product of choices made in the research process" [3]. Active productions of ignorance come in a number of flavors, from strategic ignorance to obscurantism, and other anti-epistemic strategies (creating doubt where knowledge exists) to virtuous ignorance.

**Table 1.** Typologies of ignorance.

| | TOTAL IGNORANCE | | | |
|---|---|---|---|---|
| *Classic Typology* | Beyond the knowable | Within the knowable | | Beyond the knowable |
| | *Recognized Ignorance* | | *Unrecognized Ignorance* | |
| *Agnotology Typology* | Deep ignorance | Plain recognized ignorance | Plain unrecognized ignorance | Thoroughly opaque ignorance |
| | | **Strategic ignorance** | **Passive construction** | |
| | | **Virtuous ignorance** | | |
| | **Obscurantism and Anti-Epistemic Strategies** | | | |

　　Strategic ignorance (though this might be better termed strategic *ignoring*) is the intentional use of ignorance for strategic purposes. Much of the focus on strategic ignorance has been on less than admirable instances. Indeed, McGoey, in her landmark book on the topic, defines strategic ignorance as "actions which mobilize, manufacture or exploit unknowns in a wider environment to avoid liability for earlier actions" [23]. This includes examples of bank executives ignoring their underlings' actions so as to able to truthfully claim ignorance of them. Strategic ignorance is related to what DeNicola terms "nescience", "what we or others have determined not to know" [24]. In fact, DeNicola considers strategic ignorance as a form of "nescience". Others forms include *rational ignorance* (when knowing is not worth the effort) and *willful ignorance* (when it is better not to know some piece of information, especially where knowing such a piece of information can be painful or paralyzing, for example, "when one is unable to summon the courage to jump a ravine and thereby get to safety, because one knows that there is a serious possibility that one might fail to reach the other side" [25]). As these are closely related, I will stick with strategic ignorance as an overarching term.

　　Virtuous ignorance is a special case of active construction where knowledge is intentionally not pursued for moral reasons. A classic case we may think of is the near universal decision to ban human cloning. Banning human cloning means we remain ignorant of many of the things we could and would learn through the development and application of this technology. Note that some scholars have argued this intentional creating of ignorance is not virtuous [26]. Still, broadly, the consensus view is that it is better for us to remain ignorant than to start playing around with human cloning. It is a virtuous ignorance.

　　Both strategic ignorance and virtuous ignorance require intentional ignorance—actively not-knowing something that could be known. As such, they are a form of *recognized ignorance* but not *deep ignorance* (or at least, they do not necessitate *deep ignorance*). In both cases, we recognize we are ignorant and we could, in principle, gain knowledge about the issue, but we choose not to. Obscurantism, on the other hand, while also a form of active ignorance, is one that plays on the intersection between plain old *recognized ignorance* and *deep ignorance*.

Obscurantism and what Carrier terms "anti-epistemic strategies" are the moves made by various epistemic actors that "damage or hurt the production of knowledge" [27]. Classic examples include Wakefield's false MMR vaccine study, and the well-rehearsed doubt-creating moves of both the tobacco lobby and the climate-change denial lobby. These actors have, through various means, created doubt and uncertainty, and have actively worked to maintain a level of ignorance, most classically by using the existing knowledge-creating structures ("we need more research still before we can be certain humans cause climate change", they might say). The doubt-flavored variety of ignorance that emerges relies, in part, on balancing what we do not yet know and what we cannot yet know. In doing so, anti-epistemic strategies, while actively creating, fostering, and feeding ignorance, straddle the space between *recognized ignorance* and *deep ignorance*.

The passive construction of ignorance, on the other hand, is the residual ignorance of our epistemic efforts and sits squarely within the realm of *unrecognized ignorance*. The manner in which knowledge is pursued always leaves some areas non-investigated. An exemplar case is the standard use of crash-test dummies that are sized to represent male drivers as opposed to female drivers [28]. As a result, car safety engineers are significantly more ignorant of the effects of crashes on female drivers than on male drivers, and are therefore less able to optimize car safety for all genders equally. The very real-world result of this ignorance is that female drivers are significantly more likely to suffer injury as a result of accidents. Indeed, "the odds of a belt-restrained female driver sustaining an MAIS 3+ and MAIS 2+ injury were 47% (95% CI = 27%, 70%) and 71% (95% CI = 44%, 102%) higher, respectively, than those of a belt-restrained male driver" [28].

The passive construction of ignorance, then, is ignorance of some fact that could be known, but the lack of knowledge has not been noticed. As such, the passive construction of ignorance is a form of *unrecognized ignorance*, but not of *thoroughly opaque ignorance* (we could, in principle, gain knowledge about the issue). Of course, the catch with providing an example, such as the effect of gender assumptions in crash-test dummies, as I have above, is that to provide an example requires recognizing the ignorance in the first place. Once this happens, the ignorance moves from unrecognized passive construction to recognized ignorance. One can only hope that as this has been recognized, steps will likely be taken to redress the ignorance.

What stands out from these distinctions within ignorance, and the discussions motivating the typologies, is that aside from virtuous ignorance (which is noted, but rarely discussed in more depth), ignorance is a bad object. It is something to be either managed or overcome. This is particularly true in science where it has been noted that "uncertainty is there to be banished, and ignorance is to be rolled back beyond the horizon" [5].

## 4. Ignorance in Science

The negative aspects of ignorance have been well-rehearsed—from the negative effects of false beliefs, be it intentional misinformation or otherwise, to the harms of unrecognized ignorance, from naïve p-hacking to biases such as the case of crash-test dummies. I therefore will not discuss them further here but take them as acknowledged. What has received less, if any attention, are the many very important and positive roles and functions ignorance plays in science. Ignorance, I will suggest, is not only beneficial to science, but is in fact an essential component of science. I will focus on three aspects. Firstly, ignorance as a foundational basis for knowing. That is the easy one. Secondly, ignorance as central to the culture of science as is standard practice. Lastly (and more importantly), ignorance as a fundamental and powerful tool essential to create increasingly important, insightful, and complex knowledge. Science not only interacts with ignorance, it intentionally uses ignorance as a tool to improve its epistemic pursuits.

Ignorance is a pre-requisite in science. Science, no matter how we interpret the term, is at least to some extent, concerned with gaining new knowledge and improving our understanding [29]. Of course, there would be no need to improve our understanding or to gain new knowledge if we were not, to begin with, ignorant. It takes not knowing

something for that knowledge to be novel. Scientific discoveries can only be "discovered" if they are not already known—if we are currently ignorant of them. Some science will lead to new knowledge accidently—scientists might sort of stumble on something new. However, much of science is about intentionally responding to an acknowledged lack of knowledge, an acknowledged ignorance. Highlighting what is not known but important or relevant to know in a field is often the first step in science. Indeed, the recent moves to increasingly pre-register [30,31] trials and studies rely, amongst other things, on first clearly acknowledging and articulating our ignorance (and how we plan to overcome it) before we even start our investigations. Ignorance, then, is the first step to science. Firestein, in fact, goes as far as to argue that ignorance is "the most critical part of the whole operation" (of science) [6], especially *recognized ignorance*.

Ignorance also plays a critical role in the cultural practice of science. While science can be defined in terms of method or in terms of sets of fact [32], science is also a set of cultural norms and practices. The culture of science includes everything from the way citations are used as a marker of expertise [33] to the value put on professional development activities beyond lab-based skills [34]. A fundamental cultural practice in science is peer review. Indeed, for a field fundamentally concerned with producing high-quality knowledge, "peer review is the principal mechanism for quality control in most scientific disciplines" [35]. In fact, peer review not only acts to ensure the quality of the work, but is also perceived by most academics to improve the quality of their publication, where "double-blind peer review is considered the most effective form of peer review" [35]. Blinding, fundamentally, is about intentionally creating ignorance. Creating ignorance in the reviewer about who authored the piece and ignorance in the author about the reviewer aims to reduce bias in the reviewer and minimize retributions or repayments of favors. A blinded peer review creates the kind of (presumably) beneficial ignorance commonly seen in settings that deal with justice and fairness, such as ignoring facts unrelated to a case in jury settings or Rawls' veil of ignorance in political theory [36]. In doing so, blinded peer review uses a form of active construction of ignorance to enable science to better achieve its epistemic aims.

Lastly, ignorance is one of the most powerful tools we have in our knowledge-making activities, and especially in science. Much of science is about focusing on the specific issue and topic under investigation. Indeed, to glean evermore in-depth, insightful knowledge, distracting noise needs to be removed. Consider the case of models, one of science's most powerful tools. Models are always a simplified version of the real world (the only full and complete model of the real world is the real world itself), so modelers must decide which variable to include in a model, and which to leave out. When modelling the movements of a given bird species in a rural setting, the modeler has to decide how much detail to include (e.g., with regard to villages, are the church bells towers included? Is the ringing of church bells included?, etc.). Whichever choice is made, some aspects will not be included. Indeed, the power of models stems from the clarity that comes from focusing on only what matters to the question at hand (and extrapolating from there). Likewise, consider the case of control in experiments. Control in experiments—understood here as the efforts taken to minimize the effects of all variables except the variable under observation—are a fundamental aspect of science [37]. Such form of control in experiments is, fundamentally, about ensuring aspects not under investigations do not interfere with the experiment. In effect, such control measures in experiments require we ignore the variability found in the real world. Both models and controls in experiments necessarily require ignoring—ignoring all the aspects that are considered tangential and unrelated to the object of study. This requires a form of *strategic ignorance*. This *strategic ignorance* also leads to a *passive construction of ignorance* because ignoring aspects considered unrelated leads to ignorance of the effects and interactions with and between these left-out variables. While this can sometimes lead to shortcomings in scientific understanding (perhaps the church bells in our model does affect the movement of some birds), by and large, the use of ignorance in models and in controls is what allows science to provide increasingly precise, rigorous knowledge.

(Strategic) ignorance, it turns out, is one of science's most powerful and important tools. Yet ignorance is rarely the subject of science communication.

## 5. Making Science Public

Science communication's ignorance of ignorance flags an important shortcoming for public understanding of science. Science communication in all its guises, whether concerned with science literacy and attitudes to science, or concerned with engagement with science and participation in science [38], is underpinned by profoundly social and political concerns. Most notable among these is the view that science is an essential aspect our everyday lives and our democratic decision making. Indeed, issues from climate change to COVID-19 vaccines highlight the importance of science in both individual (e.g., to drive an electric car or not) and political decisions (e.g., which party to vote for), and hence to a well-functioning society and democracy. This centrality of science in our social and political lives gives reason to the view that there is "knowledge with which everybody ought to be familiar" (Bauer, Allum, and Miller, 2007). While the transfer of knowledge is fundamental to all movements in science communication, there are important questions about what kind of knowledge we are communicating, and what kind of knowledge we should be communicating [39].

*Critical Science Literacy*

Despite scholarship arguing for ever more participatory forms of science communication, the practice of science communication seems to still predominantly favor one-way, literacy-focused communication practices [40]. Furthermore, an increasing knowledge of scientific facts and methods is important. Indeed, both textbook scientific facts, and to a lesser extent scientific methods and processes, have long been the standard object of communication. However, science is more than facts. Science arises from a rich, complex, socio-cultural set of practices—from the peer-review process to the role of models—equally important to the making of scientific knowledge as the scientific method. These more nebulous, socio-cultural aspects and processes with which scientific facts and methods get created and become accepted have not, however, been the recipients of as much love from science communicators as the facts of science have. Yet understanding science without understanding the processes that makes science what it is seems nigh impossible. More troubling, without an understanding of these socio-cultural aspects and processes, accepting the scientific facts requires blind faith. Responding to this challenge, there have been calls to move away from primarily (and almost solely) communicating textbook facts, what we might call "classical science literacy", towards what has been termed "critical science literacy" [41]. Similar to classical science literacy, critical science literacy has in its sights increasing awareness and knowledge, but what differs is what that knowledge is. Proponents of critical science literacy argue if we truly want a public that understands science such that the public can meaningfully engage with the science relevant to their everyday and political decisions, we need to increasingly make knowledge about the culture of science public, "the kind of everyday, tacit knowledge of "how things work" that members of a culture take for granted but outsiders can find mystifying" [7]. For proponents of critical science literacy, simply knowing textbook facts is insufficient for evaluating the validity and robustness of scientific claims and insufficient for truly engaging with the ideas and debates.

Being able to assess, evaluate, and intellectually engage with science, especially in the current internet-driven information-rich climate, is essential, or "at least it should be considered an essential component of science literacy" [41]. As a move in science communication, critical science literacy continues on the path that claims scientific knowledge is central to our everyday life, core to our decision making, and that everyone ought to have either knowledge of or access to some fundamental scientific basis for such decisions. However, it argues that in a world where we face information over-saturation, the capacity of individuals to understand, assess, and make sense of science and scientific claims is one of the most important aspects of science literacy. Fundamentally, critical science literacy is

about increasing knowledge in the form of skills rather than fact, specifically, "the skills needed by journalists [and others] to recognize scientific legitimacy and appropriately represent scientific claims" [7].

Here, we come back to ignorance, or more precisely, to the importance of the public understanding of ignorance in science. As Priest argues, the philosophical and sociological underpinnings of scientific practices and processes are as much in need of being made public and explicit as the content of scientific claims (Priest, 2013). Understanding both the good and bad of ignorance in science—being able to tell when it is appropriately used and when it is misused or abused—is essential "to recognize scientific legitimacy and appropriately represent scientific claims", as Priest puts it. Ignorance, as shown above, is intimately linked to science. However, ignorance is also often perceived as almost diagonally opposite to science. Science is knowledge; ignorance is its absence. As observed by Ravetz, ignorance (other than being formalized in terms of risk, uncertainty, and probabilities) is largely ignored in science: "training in science and scientific research systematically fosters ignorance of ignorance", he tells us [5]. As a result, the public communication of ignorance in science—or making ignorance public—does not even make it on the radar of those advocating for or working towards a great public understanding of science. Yet, it should.

In this context, when I argue for making ignorance public, what I want to suggest is that we make *the use* of ignorance more explicit. We rehabilitate ignorance as an important aspect and tool in science. This relates to the skills Priest [7] discusses as needed to recognize and understand science. These skills might include an appreciation of what kind of ignorance is an appropriate target for scientific ignorance, namely *plain recognized ignorance* as opposed to *deep ignorance*, and an understanding of the use of ignorance as a tool in generating knowledge, such as the use of *strategic ignorance*, say in the building of models or the design of an experiment. Such knowledge is central to critical science literacy, both in terms of determining when a scientific claim is legitimate, and when a claim is not legitimate, whether that illegitimacy is naïve (say a straight forwardly false claim) or intentional (such as one driven by an obscurantism or anti-epistemic strategy). The making public of ignorance on specific claims or facts is more complex, and is beyond the scope of this specific paper.

Before wholeheartedly advocating for a public understanding of ignorance (in science), there is a reservation, an objection, that is important to consider. The concern here is that, given what seems like an already significant push-back against science (at least on some fronts), highlighting ignorance—highlighting what is not known, rather than what is known—might lead to more distrust in science, or worse still, give fodder to those fostering doubt in science. Indeed, this may seem intuitively correct, but there are empirical as well as moral reasons to overrule this objection. Empirically, this concerns rests on an assumption that if we were to withhold from making ignorance public, then the concerns over ignorance as a reason to distrust science would be significantly lessened. This seems to ignore much of the socio-political setting which drives much of this distrust. In the environmental sector at least, much of the claims that foster distrust have come out of well-funded, well-researched anti-epistemic campaigns, irrespective (and often intentionally dismissive) of what standard science communicators might say [42]. Not making ignorance public in the hope that it will not make it onto the radar of those potentially distrusting science is a false hope.

There is also a moral concern with holding back on making ignorance public. As mentioned above, one of the most important motivations for making science public is to empower individuals to assess, evaluate, and intellectually engage with science, given the latter is such an essential aspect our everyday lives and our individual and democratic decision making. Science is communicated in the spirit of making knowledge public, such that individuals can make informed decisions [43,44]. If that is a central premise of making science public, then deliberately withholding information about the role of ignorance in science seems lacking in integrity, if not outright contradictory to the notion of empowering

individuals to make informed decisions. So, while it is worth considering when and how making ignorance public interacts with (dis)trust in science, given both the empirical and moral rebuttal, this concern should not be held as an objection to a greater public understanding of ignorance.

Most importantly, as mentioned above, the argument here is about the making public of ignorance as an important aspect and tool in science, not the making public of ignorance of individual claims. Indeed, far from dismissing the concerns that there are opponents to the dominant scientific views, and that we ought to carefully consider how the ignorance of the public is used by opponents to misuse and abuse science, the public understanding of ignorance that I am advocating is about building the critical skills that permits differentiating the legitimate use of ignorance, as opposed to its misuse and abuse.

Ignorance, then, far from being the epistemic villain, is very much central to how science operates, from being the source of curiosity, to the process of peer review, to being a key tool in allowing us to focus our attention, reduce noise, and increase understanding and explanatory power. Understanding the role ignorance plays in science is critical to being able to truly and meaningfully engage with science, to being able to understand the process and the limitations of science. So, if we truly mean what we say when we talk about wanting to have a greater public understanding of science, about increasing critical science literacy, then, what we need is an understanding of when and how ignorance is used in science, as well as when it is misused and abused. So, here is the challenge for those engaged in and committed to a better public understanding of science: if we want meaningful public understanding of science, then we need to foster a public understanding of ignorance.

**Funding:** This research received no external funding.

**Informed Consent Statement:** Not applicable.

**Data Availability Statement:** Not applicable.

**Conflicts of Interest:** The author declares no conflict of interest.

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
