# Peer review of "Public Understanding of Ignorance as Critical Science Literacy"

_sustainability, doi:10.3390/su14105920_

Round 1
Reviewer 1 Report
This paper is a philosophical discussion of the concept of ignorance, with special reference to ignorance in science and its potential effects on public trust or belief. It’s well written (although I will note a few minor typos below), and it raises important points, but I have a nagging concern that the word “ignorance” is perhaps being put to too much use here. Words like “uncertain” or “unknown” seem at least as relevant but less pejorative; a connotation of “ignorance,” at least in my view, is that it is subject to – and should be subject to – being rectified. Ignorance, in other words, is somehow wrong. If one is “ignorant,” perhaps one hasn’t done one’s homework and in that sense the ignorance is thus somewhat of a choice, or at least a consequence of a choice, that should not have been made. Or maybe the person was not able to learn something they should have learned; ignorant people may be seen as dull-witted in that sense. That is, we clearly cannot know what we do not know, but being ignorant implies a degree of negligence or even a defect. It is clearly pejorative; if I tell you that you are ignorant, this is clearly an insult. Perhaps it is not the fault of the ignorant person (or group) if they never had the chance to get an education in certain areas, as in “ignorant peasants” or – possibly worse – “ignorant blokes.” But if a scientist or research team or entire field is ignorant of something which is relevant to their work and which could and should have been known to them, there is a connotation that this shouldn’t be the case. The person or people involved should buckle down to work and become knowledgeable instead. The words “uncertain” or “unknown” don’t have this negative a connotation. Thus if you ask me whether “ignorance” is an essential part of science, I would very much hesitate to say “yes,” but if you ask me whether science is always “uncertain” or always involves “uncertainty,” I would definitely agree. So if I were answering a survey about my view of ignorance in science, I would have to say it is a bad thing, but I wouldn’t say that about uncertainty. Similarly for “unknown,” scientists generally seek to study questions with unknown answers, but this does not necessarily involve "ignorance." This can possibly be resolved without deep rewriting if the authors devote more attention to making it clear at the beginning that they are using "ignorance" in a very broad sense. However, I leave it to the editors to make that determination. Because I don't agree with these authors use of the very important word "ignorant" in this work, I cannot be overly enthusiastic about this otherwise interesting paper.
Minor typos I noticed:
- 6 standardly practice [s/b standard practice?]
- 7 because ? [why the ? was the author making a note to add something that is missing?]
- 8 claims.” (Priest, 2013) [punctuation, also spacing below on same page]
- 10 made public> [in bio]
Author Response
Thanks for the comments. These have been very helpful in improving the argument. I have responded to both reviewers in one table. Attached.

Reviewer 2 Report
The journal description for research essays includes an insistence on details that can be reproduced in the lab. As this submission is not about a lab experiment, I assume it fits the category of “communications.” I also assume that the words “cross-disciplinary,” “cultural,” “social sciences and humanities” fit that genre.
I was a little surprised at the number of errors and infelicities in the text, but I guess that is a problem for the proofreader.
In any case, it should be stressed that this review is not being written by a scientist but someone in the humanities interested in science communication to the public. Of course I am also a citizen in a democracy and thus regard such communication, about climate change and the pandemics certainly, as a matter of life and death not only for myself but more importantly for civilization and our species.
Hence, I agree completely that we need to stress that “science is an essential aspect our everyday lives and our democratic decision making. Indeed, issues from climate change to COVID vaccines highlight the importance of science in both individual (e.g. to drivean electriccar ornot) and political decisions (e.g. which partyto votefor), and henceto a well-functioning society and democracy.”
I can relate especially to “not knowing …. either lacking any knowledge or holding a false knowledge….the more we know we don’t know.” I liked especially Ungar on “ignorance that exists beyond the boundaries of knowledge, such as scientific ignorance about new or unknown phenomena.” I would contrast that with what the public regards as the excessive pride of “experts.”
I also liked Wilholt on unrecognized ignorance “that is not only left off our radar, but could not even make it onto our radar as we don’t have the conceptual capacity to formulate the question.” I especially liked the example of “Aristotle’s ignorance of what it’s like inside a black hole.” I am impressed that Cartesian science is giving way to science that can deal with “risk, uncertainty and probabilities.” And I liked the definitions of “intentional ignorance — actively not- knowing something that could be known,” and of “rational ignorance (when knowing is not worth the effort), and of wilful ignorance (when it’s better not to know such, especially where knowing some piece of information can be painful or paralysing, for example, ‘when one is unable to summon the courage to jump a ravine and thereby get to safety, because one knows that there is a serious possibility that one might fail to reach the other side’”
On the other hand, my research does not extend to ignorance’s role in the scientific method or even in critical science literacy except for a focus on “the capacity of individuals to understand, assess, and make sense of science and scientific claims.”
Thinking about the general public, I disagree about the danger of giving the audience excuses for maintaining their ignorance, but then I am thinking of audiences not of scientists or those friendly to science, but the kind of doubting if not hostile audiences one is most likely to face when trying to convince the public about changes in their lives required for amelioration of pandemics and climate change.
The history of rhetoric from Aristotle to social science tells me that there is indeed a great danger that this defense of ignorance will “give fodder to those fostering doubt in science.” The topic for me then is not so much “how ignorance is used in science, as well as when it’s misused and abused,” as how the ignorance of the public is used by opponents to misuse and abuse science.
I agree that the emphasis should be on “skills rather than fact,” but wonder specifically what “skills” the speaker should have to communicate to the general public?
(If I were writing this essay for readers from the humanities point of view, I would focus on the skills stressed in traditional teaching of the rhetoric of public speaking, especially as reinforced by social science, and a related set of skills involving stories and images dictated by brain scan discoveries of the relationship between what used to be called the two sides of the brain. But the readers of this essay are not going to be from the humanities and thus other reviewers will need to evaluate the impact on scientists.
Author Response

(The authors gave the same response as above.)

Round 2
Reviewer 2 Report
Ultimately, I am persuaded by the author's "false hope" argument
I am not sure why the author deleted Aristotle's name from the black hole statement. I think it is more effective to include Aristotle's name.